# Anti-Pruritic and Immunomodulatory Effects of Coix [*Coix lacryma-jobi* L. var. *ma-yuen* (Rom. Caill.) Stapf.] Sprouts Extract

**DOI:** 10.3390/ijms252111828

**Published:** 2024-11-04

**Authors:** Eun-Song Lee, Yong-Il Kim, Jeong-Hoon Lee, Jang-Hoon Kim, Yong-Goo Kim, Kyung-Sook Han, Young-Ho Yoon, Byoung-Ok Cho, Ju-Sung Cho

**Affiliations:** 1Department of Herbal Crop Research, National Institute of Horticultural & Herbal Science, Rural Development Administration, Eumseong 27709, Republic of Korea; eslee24@korea.kr (E.-S.L.);; 2Division of Animal, Horticultural and Food Sciences, Chungbuk National University, Cheongju 28644, Republic of Korea; 3Mushroom Research Division, National Institute of Horticultural & Herbal Science, Rural Development Administration, Eumseong 27709, Republic of Korea; 4Institute of Health Science, Jeonju University, Jeonju 55069, Republic of Korea; 5Brain Korea 21 Center for Bio-Health Industry, Chungbuk National University, Cheongju 28644, Republic of Korea

**Keywords:** coix sprouts extract, HMC-1 cells, IL-31 cytokine production, mast cell infiltration, prednisolone

## Abstract

This study explored the anti-pruritic and immunomodulatory effects of Coix sprouts extract, focusing on histamine release and IL-31 cytokine production in HMC-1 cells. The extract significantly inhibited both factors, indicating its potential for pruritus relief. In a pruritus induction mouse model, Coix sprouts extract outperformed prednisolone in anti-pruritus effectiveness, also improving skin lesions and inhibiting mast cell infiltration. The extract suppressed tryptase expression, reduced release, inhibited mast cell proliferation, and lowered nitric oxide production, suggesting anti-inflammatory properties. Coix sprouts extract shows promise in suppressing inflammation and pruritus, making it a valuable candidate for clinical use. Additionally, the analysis of coixol content in Coix sprouts revealed variations in growth time, indicating their potential as functional materials with anti-pruritus and immune-enhancing applications.

## 1. Introduction

The prevalence of atopic dermatitis is increasing in industrialized countries, with rates being 2–10% and 15–30% among adults and children, respectively [1,2]. Consequently, ongoing research efforts are aimed at addressing this issue. The pathogenesis of atopic dermatitis involves multiple factors, including environmental and genetic factors, the breakdown of the skin barrier causing transepidermal water loss, decreases in skin lipid components, abnormal differentiations of keratinocytes, abnormal immune responses, and opportunities for infections due to skin lesions [3,4].

The key symptom of atopic dermatitis is severe inflammation, accompanied by skin pruritus. Immunologically, this condition is characterized by the excessive infiltration of immune cells into skin lesions. This infiltration promotes the production of inflammatory cytokines, including tumor necrosis factor-alpha (TNF-α) and interleukin-6 (IL-6). Simultaneously, CD4^+^ T cells, particularly Th2, Th17, and Th22 cells, undergo biased differentiation, producing numerous inflammation-related cytokines and chemokines that further enhance the inflammatory response [5]. Notably, the promotion of the production of the representative Th2 cytokine, IL-4, stimulates B cells, leading to an increase in plasma immunoglobulin E (IgE). This, in turn, elevates skin infiltration of inflammatory immune cells, including mast cells, exacerbating pruritus and the inflammatory response [6]. Interleukin-31 (IL-31) is a cytokine that plays a critical role in the immune system and is closely associated with pruritus [7]. It is primarily secreted by T cells and, particularly, Th2 cells, and is recognized as a key mediator of pruritus in various inflammatory skin diseases.

Mast cells play a vital role in both innate and adaptive immunity for the body’s defense. However, when excessively activated, mast cells not only secrete pruritus-inducing substances, such as histamine and serotonin, but also accelerate the production of pro-inflammatory cytokines, such as TNF-α and IL-6, thereby intensifying the inflammatory response [8]. These hyperactivated mast cells primarily induce the production of Th2 cytokines, such as IL-4, and exhibit a specific response to IgE, thereby promoting hypersensitivity reactions. This makes them contributing factors in allergic skin conditions, including atopic dermatitis [9]. Consequently, the discovery of substances capable of inhibiting mast cell activation is essential for the improvement or treatment of atopic dermatitis and other allergic skin disorders.

There is growing interest in the consumption of plant-based foods for disease prevention and treatment. Among plant-based foods, sprouts have gained attention as a health food [10]. Sprouts typically refer to the young leaves of plants that have just sprouted, usually approximately 1 week after germination. At this stage, sprouts contain substantial amounts of various amino acids, enzymes, vitamins, minerals, and dietary fibers [11]. Furthermore, during the germination process, fat and calories decrease and various bioactive compounds are generated to protect the seed from external threats [12].

Coix (*Coix lacryma-jobi* L.), also referred to as adlay or Job’s tears, is a lesser-known cereal with significant culinary value in certain Asian regions [13]. Coix seeds, originating from tropical Asia, contain a significant amount of starch in their endosperm, along with polysaccharides, proteins, and seed oil, among other chemical components [14]. The anticancer effects of Coix stem and leaf extracts [15], the antidiabetic activity of Coix seed extract [16], the improvements in inflammation and oxidative stress in a rheumatoid arthritis model [17], the preventive effects on nonalcoholic fatty liver disease [18], and the effects in high-fat diet-induced mice [19] have all been explored.

The active ingredient of Coix is coixol, which suppresses nuclear transcription factor κ B, mitogen-activated protein kinases pathways, and NOD-like receptor protein 3 inflammasome activation in lipopolysaccharide-induced RAW 264.7 cells [20]. Coix sprouts have been investigated for their in vitro anti-colon cancer, antioxidant, and anti-inflammatory properties [21]. Furthermore, it has been suggested that Coix seeds may impact human immune function; in addition [22], the polyphenol extract of Coix exhibited notable hypocholesterolemic and antioxidant activities, potentially contributing to its protective effects on cardiovascular health in vivo [23].

Therefore, this study aimed to investigate the effects of Coix sprouts extracts on pruritus suppression. The secretion of histamine and interleukin (IL)-31 was examined in the human-derived mast cell line, HMC-1, and in an acute pruritus-induced animal model. Additionally, it was sought to determine whether Coix sprouts extract could enhance immune activity and to investigate the content of coixol, a bioactive compound related to this effect, to assess its potential for development as a novel functional food ingredient.

## 2. Results

### 2.1. Inhibition of Histamine and Cytokine Release in the HMC-1 Cell Line

The inhibitory effects of Coix sprouts extract on the release of histamine, a key substance known to induce pruritus, were investigated. When HMC-1 cells were treated with phorbol myristate acetate (PMA) and calcium ionophore A23187, histamine release increased significantly from 29.2 to 215.1 pg/mL (Figure 1A). However, when treated with Coix sprouts extract at all concentrations of 25 and 50 µg/mL, a significant inhibitory effect on histamine release was observed.

The inhibitory effect of the Coix sprouts extract on the generation of IL-31, a key substance in histamine-independent pruritus induction, was also examined. Treatment of HMC-1 cells with PMA and A23187 led to a significant increase in IL-31 cytokine production, from 15.2 to 56.0 pg/mL, when compared with the untreated group (Figure 1B). However, at all concentrations of 25 and 50 µg/mL of Coix sprouts extract, IL-31 cytokine production was reduced.

### 2.2. Anti-Pruritic Effects in an Acute Pruritus-Induced Mouse Model

This study aimed to investigate the effects of Coix sprouts extract on pruritus suppression. To evaluate the antipruritic effects of Coix sprouts extract at the histological level, dorsal skin tissues were stained with hematoxylin, eosin, and toluidine blue for microscopic analysis (Figure 2A). In the control group treated with compound 48/80 (C48/80), the itching agent, notable increases in epidermal thickness (red arrows) and inflammatory cell infiltration (blue arrows) were observed, indicative of aggravated tissue damage. In contrast, the group administered with Coix sprouts extract exhibited marked histopathological improvements, demonstrating its efficacy in ameliorating skin lesions.

Coix sprouts extract (20 mg/kg) and prednisolone (5 mg/kg) were orally administered to ICR mice, followed by a subcutaneous injection of the pruritus-inducing substance between the shoulders 1 h later. The results showed that the experimental group treated with compound 48/80 exhibited a significant increase in the number of scratching events (97.6/60 min) compared with the normal control group (9.2/60 min) (*p* < 0.05) (Figure 2B).

However, the experimental group treated with the Coix sprouts extract demonstrated a remarkably superior inhibitory effect on pruritus induced by compound 48/80. The number of scratching events was reduced to 63.8/60 min, representing a 34.6% reduction in scratching frequency. Furthermore, the antipruritic effect of Coix sprouts extract was higher than that of prednisolone (80.2/60 min), a well-known antipruritic drug. Further research is needed to elucidate the molecular mechanisms underlying the antipruritic effects of Coix sprouts extract.

### 2.3. Improvement of Skin Lesions and Inhibition of Mast Cell Infiltration in Histological Changes

To confirm the anti-pruritus effect of the Coix sprouts extract at the histological level, mouse skin tissue in the dorsal area was observed using H&E and toluidine blue staining. The results showed that the control group treated with compound 48/80 exhibited a significant increase in skin thickness and barrier disruption compared with the normal control group (Figure 3A). However, the group treated with Coix sprouts extract showed an improvement in skin histology compared with the control group.

Furthermore, the changes in mast cell infiltration, which was crucially involved in pruritus, were examined using toluidine blue staining and compared with the normal control group. The results showed that the group treated with compound 48/80 had a substantial increase in mast cell infiltration (Figure 3B). In contrast, the group treated with Coix sprouts extract exhibited a significant inhibition of mast cell infiltration, and the improvement was superior to that of the reference drug-treated group (prednisolone).

### 2.4. Degranulation of Mast Cells and Inhibition of Histamine Release

To investigate changes in tryptase, a biomarker of mast cell activation that plays a crucial role in pruritus, immunohistochemical staining was performed. The results showed that the control group treated with compound 48/80 exhibited an increase in tryptase expression (Figure 4). In contrast, the group treated with Coix sprouts extract showed significant inhibition of tryptase expression, and this effect was superior to that of the reference drug prednisolone (5 mg/kg).

To assess the inhibition of histamine release, which is a key mediator of pruritus, blood was collected from mice treated with compound 48/80 and serum was prepared. Serum histamine concentrations were quantified using ELISA. The results showed that histamine release significantly increased from 4.0 ng/mL to 7.5 ng/mL in the compound 48/80-treated group compared with the normal control group. However, the group treated with Coix sprouts extract exhibited a significant suppression of histamine concentration in the serum, reducing it to 3.4 ng/mL, and this effect was superior to that of the reference drug-treated group, which had a concentration of 5.6 ng/mL.

### 2.5. Inhibition of IL-31 Expression, a Biomarker of Histamine-Independent Pruritus

To assess the inhibitory effect of Coix sprout extract on histamine release, a key mediator of pruritus, blood was collected from mice treated with compound 48/80. Serum was prepared, and histamine levels were quantified using ELISA. In the compound 48/80-treated group, histamine release was significantly elevated compared with the normal control group (Figure 5A). However, the administration of Coix sprout extract significantly suppressed the increased histamine levels, demonstrating a superior effect compared with the reference drug, prednisolone.

To investigate the effect of Coix sprouts extract on IL-31, a key cytokine involved in histamine-independent pruritus, real-time PCR was conducted to measure its mRNA expression. The results showed that in the skin tissues, IL-31 mRNA expression increased significantly by 1.4 times in the compound 48/80-treated group compared with that in the normal control group. However, the group treated with the Coix sprouts extract exhibited a trend toward decreased IL-31 expression, although this reduction was not statistically significant (Figure 5B).

### 2.6. Inhibition of Tryptase and IL-31 Protein Expression

To investigate the effect of the Coix sprouts extract on the expression of tryptase and IL-31 proteins, Western blot analysis was performed. The results showed that in skin tissues, the expression of tryptase and IL-31 proteins was significantly increased in the compound 48/80-treated group compared with that in the normal control group. However, the group treated with Coix sprouts extract exhibited significant inhibition of the increased expression of tryptase and IL-31 proteins, and this effect was superior to that of the reference drug prednisolone (Figure 6).

### 2.7. Effect on Spleen Cell Proliferation

To analyze the effect of Coix sprouts extract on spleen cell proliferation, Coix sprouts extract (0, 25, 50, and 100 μg/mL) was applied at various concentrations. Mast cells were isolated from the spleen tissues of BALB/c mice. The results showed that Coix sprouts extract increased mast cell proliferation in a concentration-dependent manner, up to 100 μg/mL, and inhibited proliferation at a concentration of 200 μg/mL. Positive control substances, such as the mitogen concanavalin A and lipopolysaccharides, also increased mast cell proliferation (Figure 7A).

### 2.8. Effect on NO Production in Spleen Cells

In a preliminary study, it was found that the minimum significant inhibitory concentration of Coix sprouts extract for NO production was 25 μg/mL, and all concentrations of Coix sprouts extract showed a cell viability of over 100%, indicating no cytotoxicity.

To investigate the effect of Coix sprouts extract on NO production in spleen cells, Coix sprouts extract (0, 25, 50, and 100 μg/mL) was applied at different concentrations. After 1 h of treatment, the spleen cells were stimulated with lipopolysaccharide (LPS, 1 μg/mL) and cultured for 16 h. The amount of NO produced was determined using Griess assay. The results showed that the group treated with LPS alone exhibited a significant increase in NO production compared with the untreated group. However, the group pretreated with Coix sprouts extract showed a concentration-dependent decrease in NO production compared with the LPS-treated group (Figure 7B).

### 2.9. Effect on Cytokine Secretion in Spleen Cells

To investigate the effect of Coix sprouts extract on cytokine secretion in spleen cells, Coix sprouts extract was applied at concentrations of 0, 25, 50, and 100 μg/mL, and the cell supernatant was collected for ELISA analysis (Figure 8). The results showed that in spleen cells treated with LPS (lipopolysaccharide), Coix sprouts extract treatment led to a concentration-dependent increase in the secretion of IL-6 and tumor necrosis factor-alpha (TNF-α). However, IL-1β, IL-4, and IFN-γ levels did not increase significantly.

### 2.10. Coixol Content Analysis of Coix Sprouts

The analysis equipment utilized in this study was a liquid chromatograph, and the precision of the analysis, including the preprocessing procedures and results, was confirmed. The precision of the analysis results for each sample ranged from 0.3% to 6.3%, satisfying the food evaluation criteria, with all results below 10%. The coixol content of Coix sprouts was analyzed based on the days after sowing, separating the aboveground and underground parts, and the ratio is presented in Table 1.

As growth progressed, the total coixol content in the aboveground parts decreased. The highest content was observed on the third day (95.46 mg/g), whereas the lowest was on the ninth day (41.3 mg/g), resulting in an approximately 2.3-fold difference. The observed decrease in coixol content in the aboveground parts of Coix sprouts with advancing growth days was consistent with previous research findings [24]. The ratio of coixol content in the shoots to roots exhibited a similar decreasing trend with increasing growth.

Coixol, a useful component of Coix, belongs to the alkaloid category of natural compounds and has been reported to possess outstanding anti-inflammatory, antioxidant, and antiallergic properties [25]. Generally, secondary metabolites in plants serve as substances that protect plants against biological or abiotic stress [26]. Similar to other plants, Coix sprouts actively produce coixol during the early growth phase of young leaves germinated from seeds.

The results of this study indicate that coixol content varies with the number of growth days, with a decreasing trend as the number of growth days increases. Coixol content was highest during the early growth stage, particularly on the third day after sowing (Table 2). However, anti-inflammatory activity increased as the sprouts matured, peaking around the ninth day after sowing [24]. Additionally, experiments conducted on the ninth-day sprouts confirmed their effects in inhibiting pruritus and enhancing immune activity in this study. Based on the results of this study, it is suggested that Coix sprout extract could be utilized for the treatment of skin and inflammatory diseases. Further research and clinical trials are required to confirm these potential therapeutic effects and evaluate their applicability.

## 3. Discussion

Generally, secondary metabolites in plants serve as substances that protect plants against biological or abiotic stress [26]. Similar to other plants, Coix sprouts actively produce coixol during the early growth phase of young leaves germinated from seeds. Coixol, a useful component of Coix, belongs to the alkaloid category of natural compounds and has been reported to possess outstanding anti-inflammatory, antioxidant, and antiallergic properties [25]. To investigate the efficacy of sprouts germinated from Coix seeds, which contain a variety of functional coixol compounds, a study was conducted to examine their anti-pruritus and immune effects.

When the anti-pruritus efficacy was first investigated in the HMC-1 cell line, histamine release increased significantly when cells were treated with PMA and A23187 but was significantly inhibited by Coix sprouts extract at concentrations of 25 and 50 µg/mL. Similarly, IL-31 production, which increased with PMA and A23187 treatment, was also reduced by the extract at the same concentrations. These inhibitory effects may be beneficial in controlling skin pruritus and may be useful for the treatment of skin conditions, such as atopic dermatitis.

Atopic dermatitis is characterized by the excessive production of inflammatory cytokines, such as tumor necrosis factor-alpha (TNF-α) and IL-6, which are closely related to inflammation and pruritus. Among CD4^+^ T cells, Th2, Th17, and Th22 cells are known to develop excessively and produce inflammatory cytokines and chemokines, thereby promoting inflammatory responses [5,27]. Th2 cells produce IL-31, which induces pruritus. In addition, in chronic atopic dermatitis, excessive IL-4 production by Th2 cells stimulates B cells, leading to increased plasma IgE levels. Elevated IgE levels promote mast cell degranulation in the skin, leading to pruritus, immune cell infiltration, and the worsening of atopic dermatitis.

Atopic dermatitis is often accompanied by severe pruritus and chronic skin inflammation [3]. Therefore, effective pruritus suppression is crucial for improving atopic dermatitis [28,29].

Coix sprouts extract was found to be effective in suppressing pruritus in HMC-1 cells, and its anti-itch efficacy was also effectively demonstrated in animal models. In an animal model, treatment with Coix sprouts extract led to an improvement in skin tissue histology and the significant inhibition of mast cell infiltration. These results indicate that the Coix sprouts extract may help improve skin conditions and suppress inflammation. The Coix sprouts extract effectively inhibited tryptase expression and histamine release, which are associated with mast cell degranulation and pruritus, respectively. This indicates its potential for managing pruritus and related skin conditions, such as atopic dermatitis. Coix sprouts extract reduced the expression of IL-31, a biomarker of histamine-independent pruritus. Although this reduction was not statistically significant in this study, it suggests the potential of Coix sprouts extract in mitigating pruritus associated with conditions, such as atopic dermatitis.

The concentrations of natural product extracts employed for evaluating anti-pruritic efficacy included 25 µg/mL of mugwort distilled water extract [30], 50 mg/kg of grape branch 80% ethanol extract [31], and 50 mg/kg of goldenrod and honeysuckle 80% ethanol extract [32]. These extracts were utilized according to specific crop conditions.

In experiments involving the secretion of histamine and L-31 in the HMC-1 cell line, the anti-pruritic effect was evident at low concentrations (25 and 50 µg/mL) of Coix sprouts extract. Accordingly, an equivalent concentration of 20 mg/kg was prepared and administered in animal studies. Although the direct comparison of anti-pruritic effects with other crops may be challenging, Coix sprouts extract exhibited a significant anti-pruritic effect even at low concentrations, underscoring its potential efficacy.

When pruritus occurs, the skin is scratched with fingers, leading to damage to the epithelial cells [33]. This damage induces an inflammatory response as a defense mechanism and initiates a cycle in which various cytokines are released. Consequently, pruritus and immune responses are interrelated. When the immunostimulatory activity of crude polysaccharide isolated from wild ginseng adventitious root was evaluated, where LPS was used as a positive control and nitric oxide production was assessed, the results indicated that nitric oxide production did not exhibit significant activity at low concentrations; however, significant differences were observed at concentrations of 50 and 100 µg/mL [34]. In another study, the immune-enhancing effect of rice bran fermented with shiitake mushroom mycelium on mouse macrophages and spleen cells was investigated. The secretion levels of IL-1β, IL-10, IL-6, and TNF-α increased in a concentration-dependent manner. At concentrations above 4 µg/mL, the secretion levels were higher than those of the LPS-treated positive control group (1 µg/mL) [35]. Furthermore, when the immunomodulatory effect of red ginseng on cytokine secretion in vivo was assessed, the blood levels of TNF-α and IL-6 were elevated in both groups administered with red ginseng [36].

In spleen cells treated with Coix sprouts extract, cell proliferation and nitric oxide production increased in a concentration-dependent manner. Additionally, the secretion of IL-6 and TNF-α in spleen cells was enhanced, suggesting an increase in immune activity. These findings indicate that Coix sprouts extract may have significant potential as an industrial strategy for anti-pruritus and immune-boosting applications.

## 4. Materials and Methods

### 4.1. Experiment Materials

The Coix sprouts used in this experiment were grown under conditions of a 27 °C temperature and 98% humidity, with irrigation applied thrice, using a metal halide lamp for illumination and harvested on the 9th day after germination according to the previous study [24]. The RBC lysis buffer was purchased from BioLegend (San Diego, CA, USA). RPMI-1640 medium, penicillin–streptomycin, and fetal bovine serum were obtained from Thermo Fisher Scientific (Waltham, MA, USA). The Griess reagent, concanavalin A, and lipopolysaccharide were purchased from Sigma-Aldrich (St. Louis, MO, USA). The Quanti-MAX™ WST-8 Cell Viability Assay Kit was obtained from BIOMAX (Guri, Republic of Korea). ELISA kits for IL-6, IL-17, tumor necrosis factor-alpha (TNF-α), IL-1β, IL-4, and IFN-γ were purchased from R&D Systems (Minneapolis, MN, USA).

### 4.2. Extract Preparation

The crushed Coix sprouts were subjected to extraction by adding 70% ethanol to 40 times their weight. This extraction process was performed using a floor-standing shaking incubator (JSSI-300C, JSR, Gongju-si, Republic of Korea) at 50 °C and 300 rpm for a duration of 3 d. Following extraction, the extract was filtered thrice using filter paper and non-woven fabric. The filtered extract was then concentrated using a rotary evaporator (R-100; BUCHI, Flawil, Switzerland) and subsequently freeze-dried for preservation at −20 °C for use in the experiments.

### 4.3. Experiment Animal

The anti-itch effect of Coix sprouts extract at concentrations of 25–50 µg/mL was confirmed using the HMC-1 cell line (primary cells) in an animal model (animal test number jjIACUC-20230602-2022-0502-A1). Additionally, the extract was shown to enhance immune activity in spleen cells, the primary cells used in the study.

Four-week-old male ICR mice, raised in sterile environments, were purchased from Koatech (Pyeongtaek, Republic of Korea). They were allowed to acclimate for 1 week, during which they were provided with ample food and water. The mice were maintained on a 12 h light–dark cycle, and temperature (20–22 °C) and humidity (50–60%) were consistently controlled in their housing environment.

To investigate potential side effects, various parameters including body weight, visual inspection, and tissue-staining results were compared between the experimental and control groups. The autopsy results revealed no significant side effects, indicating the safety of the extract in the tested conditions.

#### Assessment of Anti-Pruritus Efficacy

The experimental design involved acclimated ICR mice with each group consisting of five mice. The groups were set up as follows: control, compound 48/80 (C48/80) treatment (C48/80 obtained from Sigma-Aldrich, St. Louis, MO, USA), C48/80 with Coix sprouts extract (20 mg/kg) treatment, and C48/80 with prednisolone (5 mg/kg) treatment. To measure pruritus induction and scratching behavior, stress-relieved ICR mice were individually placed in transparent acrylic cages (20 cm × 26 cm × 13 cm) for 30 min in the same experimental environment. The control group was orally administered normal saline. The experimental groups were administered the Coix sprouts extract and prednisolone, an anti-pruritus medication. Subsequently, 100 μL of C48/80 at a concentration of 50 μg/site was subcutaneously injected between the shoulder blades of the mice. The injected area was immediately recorded for 60 min using a microcamera (ONCCTV, Seoul, Republic of Korea), following the method described previously 27. The number of times mice scratched the injected area with their hind paws was counted and evaluated using a double-blind method.

### 4.4. Skin Tissue Staining

Skin tissue staining was performed using hematoxylin and eosin (H&E) stain and toluidine blue. Skin tissue samples (approximately 5 mm × 5 mm) were excised and fixed in 4% paraformaldehyde (pH 7.4). After a series of processes, paraffin blocks were prepared, and 5 μm-thick sections were cut. The tissue sections were deparaffinized and processed, followed by staining with H&E and toluidine blue. Stained sections were observed under an optical microscope (×100, Olympus, Tokyo, Japan). Immunohistochemical staining for tryptase was performed and examined under an optical microscope. Immunohistochemical staining involved the preparation of paraffin-embedded specimens using a commercially available Ready-to-use IHC kit (Biovision, Milpitas, CA, USA). The measurements were performed according to the manufacturer’s instructions.

### 4.5. Western Blot Analysis

After completion of the experiment, skin samples from the dorsal area of the mouse were obtained and stored at −80 °C for use in the experiment. Mouse dorsal skin tissues were homogenized on ice while maintaining the temperature to extract proteins. Protein quantification was performed using the Bradford protein quantification reagent. Subsequently, equal amounts of protein were loaded and separated by size using sodium dodecyl sulfate-polyacrylamide gel electrophoresis. The proteins were transferred to a polyvinylidene difluoride membrane (Bio-Rad Laboratories, Hercules, CA, USA) at 100 V for 1 h. After transfer, the membrane was blocked with 5% (*w*/*v*) skim milk in Tris-buffered saline containing 1% Tween20 (TBST) at room temperature. Following the blocking step, the membrane was washed thrice with TBST for 10 min. Subsequently, the primary antibodies (IL-31, tryptase, and actin) were applied and allowed to react at 4 °C for 16 h. The membrane was then washed thrice in TBST for 10 min. Next, secondary antibodies (goat anti-rabbit IgG HRP and rabbit anti-mouse IgG HRP) were diluted in 5% (*w*/*v*) skim milk and allowed to react at room temperature for 2 h. Following this step, the membrane was washed thrice with TBST for 10 min each in TBST. The expression levels of the target proteins were confirmed through the detection reaction of the secondary antibodies.

### 4.6. IL-31 mRNA Expression

After isolating RNA from mouse dorsal skin tissue and synthesizing cDNA, gene expression levels were analyzed using PCR. RNA was extracted from the collected tissues using RibospinTM II (Jin-ol Biotechnology, Seoul, Republic of Korea) and quantified using the Bio Spectrometer kinetic (Hamburg, Germany) with 1 μg of RNA. Subsequently, the RNA was reverse-transcribed into first-strand cDNA using the PrimeScript™ 1st strand cDNA synthesis kit (Takara Bio Inc., Otsu, Japan). PCR for IL-31 and GAPDH were performed using a thermal cycler PCR system (Bio-Rad, Inc.). The annealing temperature during PCR was set at 60 °C, and the amplification step was repeated for 30 cycles. Primers used in the experiments are listed in Table 3.

### 4.7. Measurement of Pruritus-Inducing Factors, Histamine, IL-31, TNF-α, and IL-8, in HMC-1 Cells

HMC-1 cells were plated in a six-well plate at a final concentration of 5 × 10^5^ cells per ml and cultured for 24 h in a 37 °C incubator with 5% CO_2_. After incubation, cells were treated with Coix sprouts extract at concentrations of 25 and 50 μg/mL. They were then simultaneously stimulated with the allergenic agents PMA (50 nM) and A23187 (1 μM). After 16 h, the levels of histamine and IL-31 in the cell supernatants were quantified using ELISA kits provided by Abcam (Cambridge, UK) and BioLegend (San Diego, CA, USA), respectively.

### 4.8. Isolation of Splenocytes

Four-week-old male BALB/c mice were euthanized via cervical dislocation under isoflurane anesthesia. Subsequently, the spleen was aseptically removed, minced, and then filtered through a cell strainer (40 μm) using slide glass. After filtering, red blood cells were removed using RBC lysis buffer, and the remaining cells were resuspended at a concentration of 2 × 10^6^ cells per mL in RPMI medium for use in the experiment.

### 4.9. Cell Proliferation Rate Measurement

Splenic cells were plated in 96-well plates at a density of 2 × 10^6^ cells per mL and incubated for 24 h. After incubation, they were treated with Coix sprouts extract (at concentrations ranging from 12.5 to 200 μg/mL), lipopolysaccharide (1 μg/mL), and concanavalin A (2 μg/mL) and then cultured for 3 d. Following the culture period, 10 μL of the Quanti-MAX™ WST-8 Cell Viability Assay reagent was added to each well. After 4 h, absorbance was measured at 450 nm. The proliferation rate was calculated based on the absorbance relative to that of the control group without samples.

### 4.10. Measurement of Nitric Oxide (NO) and Cytokine Production

Splenic cells were plated in 48-well plates at a density of 2 × 10^6^ cells per mL and incubated for 24 h. After the incubation, they were treated with Coix sprouts extract (at concentrations ranging from 25 to 100 μg/mL), lipopolysaccharide (1 μg/mL), and concanavalin A (2 μg/mL), and then cultured for 3 d. Following the culture period, the cell culture supernatant was collected and NO production was measured using the Griess assay. The production of various cytokines was measured using commercial ELISA kits.

### 4.11. Statistical Analysis

All experimental values were expressed as mean ± standard deviation. Statistical comparisons were performed using the SPSS Statistics 22 (IBM, Armonk, NY, USA). Comparisons between experimental groups were conducted using one-way analysis of variance, and Duncan’s multiple range test was performed as a post hoc test to determine significant differences among the experimental groups. The significance level for the tests was set at *p* < 0.05.

### 4.12. Measurement of Coixol Content According to Growth Duration

For analysis of coixol content in Coix sprouts, the Coix sprouts were grown under conditions of 27 °C temperature and 98% humidity, with irrigation applied thrice, using a metal halide lamp for illumination and harvested on the 3, 5, 7, 9, and 11th day after germination.

Coixol content in the Coix sprout extract was analyzed by partially modifying the analytical method used in previous studies [37]. The Coix sprouts were extracted by adding 70% ethanol at a ratio of 40 times the sample weight. The extraction was carried out for 3 days at 50 °C using a floor-standing shaking incubator (JSSI-300C, JSR, Gongju-si, Republic of Korea). After extraction, the resulting extract was filtered thrice using a filter paper and a paper filter. The filtered extract was concentrated using a rotary evaporator (R-100; BUCHI, Flawil, Switzerland) under reduced pressure and freeze-dried for use in the experiment.

The lyophilized Coix sprout extract powder was dissolved in methanol to achieve a concentration of 10 mg/mL. The resulting solution was then passed through a 0.45-µm PTEF filter (Gelman, New York, NY, USA). Subsequently, HPLC analysis was performed using a YMC-Triart C18 column (250 mm× 4.6 mml.D., S-5 µm, 12 nm) (YMC, Kyoto, Japan) maintained at a temperature of 40 °C. The mobile phase was composed of 0.1% formic acid in water (A) and 0.1% formic acid in acetonitrile (B). The concentration gradient was adjusted over time according to the schedule provided in Table 2. The flow rate was set at 0.8 mL/min, and the absorbance of the compounds was detected at 230 nm. Ten microliters of the sample solution were injected into the HPLC system (e2695; Waters Co., Milford, MA, USA) for analysis. To determine the content of coixol in the extract, a standard calibration curve was generated by gradually diluting the coixol standard from 1 to 200 µg/mL. The content of the extract was quantified by calculating the area-to-concentration ratio based on the calibration curve.

## Figures and Tables

**Figure 1 ijms-25-11828-f001:**
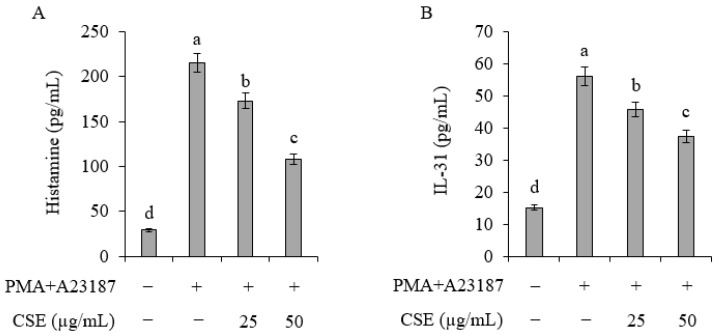
Effect of Coix sprouts extract (CSE) on histamine and IL-31 levels in HMC-1 human mast cells stimulated with PMA and A23187. (**A**) effect of Coix sprouts extract (CSE) on histamine in HMC-1 human mast cells; (**B**) effect of Coix sprouts extract (CSE) on IL-31 in HMC-1 human mast cells. The bar represents the standard error (*n* = 3). Same letters are not significantly different by Duncan’s multiple range test (DMRT) and *p* < 0.05.

**Figure 2 ijms-25-11828-f002:**
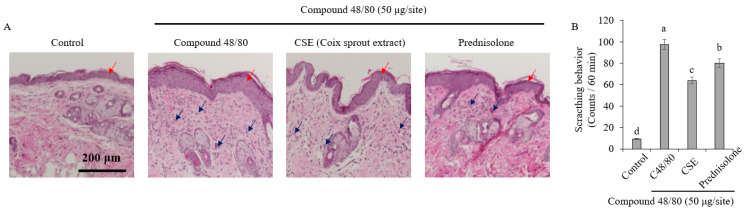
Anti-pruritic effects of Coix sprouts extract (CSE) on the histological changes and scratching behavior induced by compound 48/80 in ICR mice. (**A**) effect of Coix sprouts extract (CSE) on histological changes; (**B**) effect of Coix sprouts extract (CSE) on scratching behavior. Red arrows are pointing to the epidermal thickness and blue arrows are pointing the inflammatory cell infiltration. Bars represent standard error (*n* = 5). Means with different superscripts in the same column are significantly different (*p* < 0.05) according to Duncan’s multiple range test. Prednisolone was used as a positive control.

**Figure 3 ijms-25-11828-f003:**
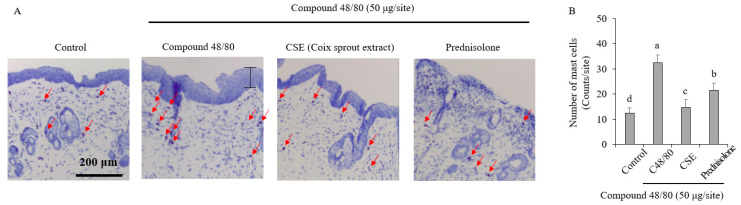
Anti-pruritic effects of Coix sprouts extract (CSE) on inhibitory effect of mast cell infiltration induced by compound 48/80 in ICR mice. (**A**) effect of Coix sprouts extract (CSE) on inhibitory effect of mast cell infiltration; (**B**) number of mast cells. Red arrows are pointing to the mast cells and black lines are pointing to the skin thickness. Bars represent standard error (*n* = 3). Means with different superscripts in the same column are significantly different (*p* < 0.05) according to Duncan’s multiple range test. Prednisolone was used as a positive control.

**Figure 4 ijms-25-11828-f004:**
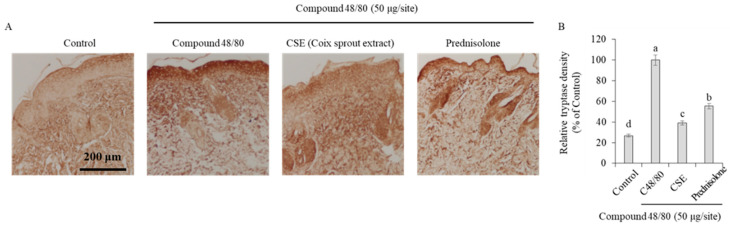
Anti-pruritic effects of Coix sprouts extract (CSE) on inhibitory effect of mast cell degranulation and tryptase density induced by compound 48/80 in ICR mice. (**A**) effect of Coix sprouts extract (CSE) on inhibitory effect of mast cell degranulation; (**B**) relative tryptase density. The brown-stained areas in the figure indicate the expression of tryptase. Bars represent standard error (*n* = 3). Means with different superscripts in the same column are significantly different (*p* < 0.05) according to Duncan’s multiple range test. Prednisolone was used as a positive control.

**Figure 5 ijms-25-11828-f005:**
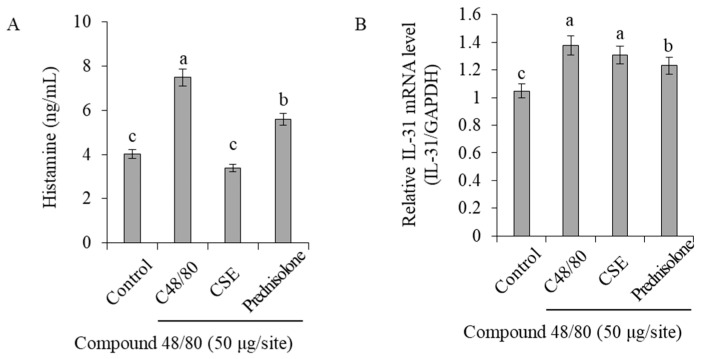
Anti-pruritic effects of Coix sprouts extract (CSE) on inhibitory effect of histamine release and IL-31 mRNA expression induced by compound 48/80 in ICR mice. (**A**) effect of Coix sprouts extract (CSE) on inhibitory effect of histamine release; (**B**) effect of Coix sprouts extract (CSE) on inhibitory effect of relative IL-31 mRNA expression level. Bars represent standard error (*n* = 3). Means with different superscripts in the same column are significantly different (*p* < 0.05) according to Duncan’s multiple range test. Prednisolone was used as a positive control.

**Figure 6 ijms-25-11828-f006:**
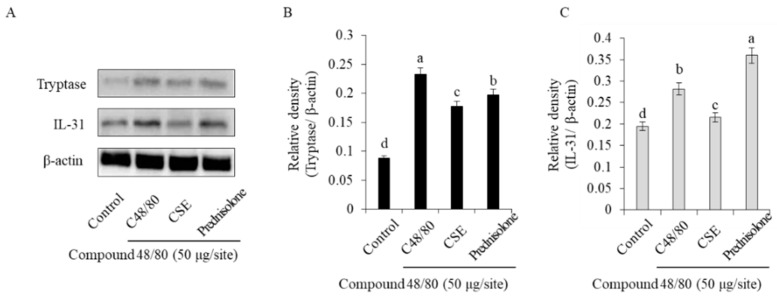
Compound 48/80 inhibitory effect of Coix sprouts extract (CSE) on tryptase and IL-31 protein expression in induced skin pruritus model. (**A**) effect of Coix sprouts extract (CSE) on tryptase and IL-31 induced by compound 48/80 in ICR mice; (**B**) relative density of tryptase/β-actin; (**C**) relative density of IL-31/β-actin. Bars represent standard error (*n* = 3). Means with different superscripts in the same column are significantly different (*p* < 0.05) according to Duncan’s multiple range test. Prednisolone was used as a positive control.

**Figure 7 ijms-25-11828-f007:**
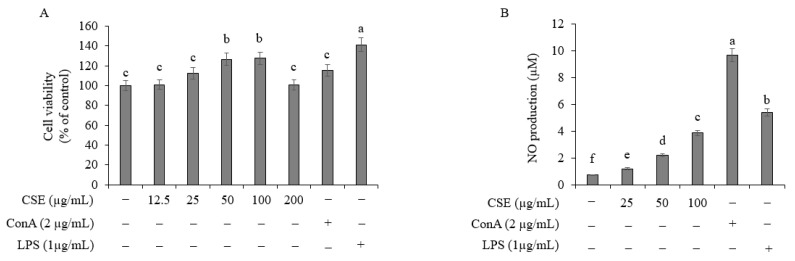
Effect of Coix sprouts extract (CSE) on cell viability and NO production in mouse-derived spleen cells. (**A**) effect of Coix sprouts extract (CSE) on cell proliferation in mouse-derived spleen cells; (**B**) effect of Coix sprouts extract (CSE) on NO production in mouse-derived spleen cells. Bars represent standard error (*n* = 3). Means with different superscripts in the same column are significantly different (*p* < 0.05) according to Duncan’s multiple range test. (−); untreated experiment, (+); treated experiment. (Con A); concanavalin A as T cell mitogen, (LPS); lipopolysaccharide as B cell mitogen.

**Figure 8 ijms-25-11828-f008:**
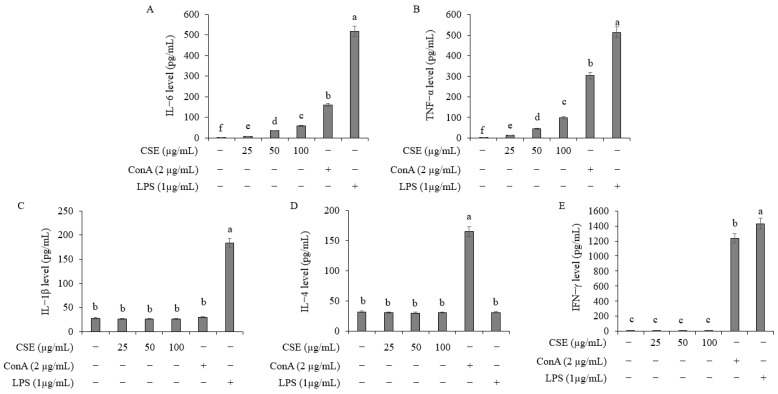
Effect of Coix sprouts extract (CSE) on cytokine secretion ability in mouse-derived spleen cells. (**A**) effect of Coix on IL-6 level in mouse-derived spleen cells; (**B**) effect of Coix sprouts extract (CSE) on tumor necrosis factor-alpha (TNF-α) level in mouse-derived spleen cells; (**C**) effect of Coix sprouts extract (CSE) on IL-1β level in mouse-derived spleen cells; (**D**) effect of Coix sprouts extract (CSE) on IL-4 level in mouse-derived spleen cells; (**E**) effect of Coix sprouts extract (CSE) on IFN-γ level in mouse-derived spleen cells. Bars represent standard error (*n* = 3). Means with different superscripts in the same column are significantly different (*p* < 0.05) according to Duncan’s multiple range test. (−); untreated experiment, (+); treated experiment. (Con A); concanavalin A as T cell mitogen, (LPS); lipopolysaccharide as B cell mitogen.

**Table 1 ijms-25-11828-t001:** Coixol content in Coix sprouts according to growth duration.

Days After Sowing	Total Coixol (mg/g, Dry Weight)	Shoot/Root Ratio
3	95.46 ± 3.74 ^a^	1.08 ± 0.02 ^ab^
5	79.74 ± 3.13 ^b^	1.14 ± 0.02 ^a^
7	71.72 ± 2.44 ^c^	1.11 ± 0.01 ^a^
9	41.30 ± 0.65 ^e^	0.64 ± 0.02 ^b^
11	47.12 ± 1.93 ^d^	0.56 ± 0.04 ^c^

Data represent the means ± SD (*n* = 3). Note: Means separation within columns by Duncan’s Multiple Range Test (DMRT, *p* < 0.05).

**Table 2 ijms-25-11828-t002:** Analytical conditions of HPLC for analysis of Coix sprout extract.

Parameters	Condition		
Column	YMC-Triart C18 (250 mm × 4.6 mml.D., S-5 μm, 12 nm)		
Detection	230 nm		
Flow rate	0.8 mL/min		
Column temp.	40 °C		
Solvent A	0.1% formic acid in water		
Solvent B	0.1% formic acid in Acetonitrile		
Gradient	Time (min)	% A ^1^	% B ^2^
	2	100	0
	45	50	50
	50	5	95
	55	5	95
	55.1	100	0
	60	100	0

(^1^) 0.1% formic acid in water; (^2^) 0.2% 0.1% formic acid in acetonitrile.

**Table 3 ijms-25-11828-t003:** Primer sequence and sequence number information.

Primer	Sequence (5′→3′)	Sequence Number
IL-31 (Forward)	CCTACCCTGGTGCTGCTTTG	1
IL-31 (Reverse)	CTGACATCCCAGATGCCCTGC	2
GAPDH (Forward)	GGCTACACTGAGGACCAGGT	3
GAPDH (Reverse)	TCCACCACCCTGTTGCTGTA	4

## Data Availability

Data are contained within the article.

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
