# Peer review of "Anti-Pruritic and Immunomodulatory Effects of Coix [Coix lacryma-jobi L. var. ma-yuen (Rom. Caill.) Stapf.] Sprouts Extract"

_ijms, 2024, doi:10.3390/ijms252111828_

Round 1
Reviewer 1 Report
Comments and Suggestions for Authors
The article 'Anti-pruritic and Immunomodulatory Effects of Coix [Coix lac-
ryma-jobi L. var. ma-yuen (Rom. Caill.) Stapf.] Sprouts Extract' is well written and contains a strong methodology. Nevertheless, it has some inaccuracies:
1). In the introduction, there is a lack of an explanation of the role of IL-31, the concentration of which was also determined.
2). 4.11. It should be 'Statistic analysis' instead of 'Extract preparation'
3). The abbreviation PMA, A23187 and compound 48/80 should be explained
4). 2.5. If a figure for histamine is included, then the results for histamine should be added.
5). 4.1. Was it determined IL-17? It is mentioned in the Materials and Methods section.
Author Response
Thank you for reviewing our manuscript.
Below is the response of our authors to your review comments.
Comments 1 : In the introduction, there is a lack of an explanation of the role of IL-31, the concentration of which was also determined.
Response 1 : We completely agree with the part where the description of IL-31 is insufficient in the introduction, and we have added references to the necessary parts and written it.
"Interleukin-31 (IL-31) is a cytokine that plays a critical role in the immune system and is closely associated with pruritus. It is primarily secreted by T cells, also, particularly Th2 cells, and is recognized as a key mediator of pruritus in various inflammatory skin diseases."
The first paragraph on page 2 of the revised paper is written in red.
Comments 2 : 4.11. It should be 'Statistic analysis' instead of 'Extract preparation'
Response 2 : Thank you for pointing out the mistake we made in our manuscript. We have corrected the red text on page 16.
Comments 3 : The abbreviation PMA, A23187 and compound 48/80 should be explained
Response 3 : We have added descriptions of phorbol myristate acetate (PMA) and calcium ionophore A23187, which were used as allergenic agents, to our revised paper (pp. 2 and 16 of the revised paper). Similarly, we have added abbreviations and descriptions of compound 48/80 (C48/80), which was used as the itching agent (page. 4).
Comments 4 : 2.5. If a figure for histamine is included, then the results for histamine should be added.
Response 4 : An important clarification was missing, and we thank you for pointing it out. We have added the results for histamine release that were missing from Figure 5A to page 7, in red.
"To assess the inhibitory effect of Coix sprout extract on histamine release, a key mediator of pruritus, blood was collected from mice treated with Compound 48/80. Serum was prepared, and histamine levels were quantified using ELISA. In the Compound 48/80-treated group, histamine release was significantly elevated compared to the normal control group (Figure 5A). However, administration of Coix sprout extract significantly suppressed the increased histamine levels, demonstrating a superior effect compared to the reference drug, prednisolone."
Comments 5 : 4.1. Was it determined IL-17? It is mentioned in the Materials and Methods section.
Response 5 : Thank you for pointing this out. We have removed the information about IL-17, which was not used as an experimental material.
Reviewer 2 Report
Comments and Suggestions for Authors
This study highlights the anti-pruritic and immunomodulatory effects of Coix sprouts extract using both animal model and cell line. The manuscript is well-structured with updated reference. The manuscript experimental design is apropriate with reproducible results based on the details given in the method section.
Major comments:
1- Figures: Most figures are not clear and have multiple faults, for example:
- Fig 2A, where are the histological and pathological features that authors assessed? please point them out on the figure or mention them in the figure legend.
- Figure 3A, Please add to the legend that red arrows are pointing mast cells. Also, authors mentioned in the text that skin thickness is different between controlled treated group, while nothing highlighted in the legend about this point.
Figure 4A, again what is the key for the histological assessment here. I guess the brown color is indicating the Tryptase expression, just add that to the legend.
-Figure 7A, B: according to methodology, all cells treated first with CSE different doses then stimulated with CON-A and LPS cultured for 3d or 16h. In the figure, authors should mark + next to CON-A and LPS for all conditions except the first one.
- Next figure which should be figure 8 not 7: A, B, C, D, and E have the same fault of the previous figure.
In Method section:
- 4.11 Extract preparation... should be statistical study.
Author Response
Comments 1 : Fig 2A, where are the histological and pathological features that authors assessed? please point them out on the figure or mention them in the figure legend.
Response 1 : Thank you for pointing out that there were parts of the figures in the manuscript that were difficult to read and difficult for readers to understand. Our authors wholeheartedly agree with the points you have made. In Figure 2A, epidermal thickness is indicated by red arrows, and inflammatory cell infiltration is indicated by blue arrows. An explanation for this is written in red on page 4 of the revised paper.
Comments 2 : Figure 3A, Please add to the legend that red arrows are pointing mast cells. Also, authors mentioned in the text that skin thickness is different between controlled treated group, while nothing highlighted in the legend about this point.
Response 2 : Thank you for pointing out the parts of the figure that were lacking in explanation. The explanation that was lacking in Legend was added in red letters and written on page 5. Also, the skin tissue thickness was marked with a black line in Figure 3.
Comments 3 : Figure 4A, again what is the key for the histological assessment here. I guess the brown color is indicating the Tryptase expression, just add that to the legend.
Response 3 : Thank you for pointing out the parts of the figure that were lacking in explanation. The part dyed brown indicates Tryptase expression, and was added to the legend on page 6 of the revised paper.
Comments 4 : Figure 7A, B, according to methodology, all cells treated first with CSE different doses then stimulated with CON-A and LPS cultured for 3d or 16h. In the figure, authors should mark + next to CON-A and LPS for all conditions except the first one. Next figure which should be figure 8 not 7: A, B, C, D, and E have the same fault of the previous figure.
Response 4 : Thank you for reviewing my manuscript. In Figure 7 and 8, Cell growth and NO production were observed by treating with Coix sprout extract alone, and ConA and LPS were used as positive controls.
Comments 5 : 4.11 Extract preparation... should be statistical study.
Response 5 : Thank you for pointing out the part where the authors made a mistake. We have corrected the incorrectly written part.
Round 2
Reviewer 2 Report
Comments and Suggestions for Authors
Authors edited the manuscript accordingly